# Refining the Role of Pyruvate Dehydrogenase Kinases in Glioblastoma Development

**DOI:** 10.3390/cancers14153769

**Published:** 2022-08-02

**Authors:** Claire M. Larrieu, Simon Storevik, Joris Guyon, Antonio C. Pagano Zottola, Cyrielle L. Bouchez, Marie-Alix Derieppe, Tuan Zea Tan, Hrvoje Miletic, James Lorens, Karl Johan Tronstad, Thomas Daubon, Gro Vatne Røsland

**Affiliations:** 1IBGC, UMR5095, CNRS, University of Bordeaux, F-33000 Bordeaux, France; claire.larrieu@ibgc.cnrs.fr (C.M.L.); antonio.paganozottola@ibgc.cnrs.fr (A.C.P.Z.); cyrielle.bouchez@ibgc.cnrs.fr (C.L.B.); gro.rosland@uib.no (G.V.R.); 2Department of Biomedicine, University of Bergen, 5009 Bergen, Norway; simon.storevik@gmail.com (S.S.); hrvoje.miletic@uib.no (H.M.); james.lorens@uib.no (J.L.); karl.tronstad@uib.no (K.J.T.); 3Department of Medical Pharmacology, Bordeaux Hospital, F-33000 Bordeaux, France; joris.guyon@u-bordeaux.fr; 4BPH, U1219 INSERM, University of Bordeaux, F-33000 Bordeaux, France; 5Animal Facility, Campus Talence, University of Bordeaux, F-33600 Pessac, France; marie-alix.derieppe@u-bordeaux.fr; 6Cancer Science Institute of Singapore, National University of Singapore, Singapore 117599, Singapore; csittz@nus.edu.sg; 7Genomics and Data Analytics Core (GeDaC), Cancer Science Institute of Singapore, National University of Singapore, Singapore 117599, Singapore; 8Department of Pathology, Haukeland University Hospital, 5009 Bergen, Norway; 9Department of Oncology and Medical Physics, Haukeland University Hospital, 5009 Bergen, Norway

**Keywords:** pyruvate dehydrogenase kinases, DCA, glioblastoma, invasion, lactate

## Abstract

**Simple Summary:**

Finding new treatment strategies is an urgent need in cancer medicine, especially for the fast-growing and aggressive primary brain tumor glioblastoma (GB). Patients with GB face a median survival of about 15 months, despite tumor resection and radio-chemotherapy. In this study, we targeted cancer cell metabolism, specifically the pyruvate dehydrogenase (PDH) regulators PDH kinases (PDHK), to disturb GB progression. Indeed, PDHK1 and PDHK2 were found to be highly expressed in GB tissue cohorts and presented different expression patterns in distinct areas of the tumor. We used patient-derived stem-like cells to knockout PDHK1 or PDHK2, and dichloroacetate (DCA) to inhibit all PDHK activity. DCA showed massive in vitro effects on proliferation and invasion but no in vivo benefit as monotherapy when injected into GB-bearing mice. Finally, mice implanted with PDHK KO cells presented a higher survival rate than the control group; this effect was enhanced by performing cranial irradiation.

**Abstract:**

Glioblastoma (GB) are the most frequent brain cancers. Aggressive growth and limited treatment options induce a median survival of 12–15 months. In addition to highly proliferative and invasive properties, GB cells show cancer-associated metabolic characteristics such as increased aerobic glycolysis. Pyruvate dehydrogenase (PDH) is a key enzyme complex at the crossroads between lactic fermentation and oxidative pathways, finely regulated by PDH kinases (PDHKs). PDHKs are often overexpressed in cancer cells to facilitate high glycolytic flux. We hypothesized that targeting PDHKs, by disturbing cancer metabolic homeostasis, would alter GB progression and render cells vulnerable to additional cancer treatment. Using patient databases, distinct expression patterns of PDHK1 and PDHK2 in GB tissues were obvious. To disturb protumoral glycolysis, we modulated PDH activity through the genetic or pharmacological inhibition of PDHK in patient-derived stem-like spheroids. Striking effects of PDHKs inhibition using dichloroacetate were observed in vitro on cell morphology and metabolism, resulting in increased intracellular ROS levels and decreased proliferation and invasion. In vivo findings confirmed a reduction in tumor size and better survival of mice implanted with PDHK1 and PDHK2 knockout cells. Adding a radiotherapeutic protocol further resulted in a reduction in tumor size and improved mouse survival in our model.

## 1. Introduction

Reprogramming energy metabolism has become a recognized hallmark of cancer [1]; therefore, therapy targeting metabolic activity is emerging. Metabolic adaptation during tumor growth is crucial to avoid cell exhaustion inducing apoptosis or cell death. Elevated glucose uptake and increased glycolytic activity, also in the presence of oxygen, are observed in most types of cancers and are considered a key characteristic of cancer cells [2]. In cells with high glycolytic flux, pyruvate is primarily fermented into lactate and NAD+ by the enzyme lactate dehydrogenase (LDH). Lactate is then transported outside the cells by the monocarboxylate transporters (MCTs/SLC16), resulting in a decrease in the pH of the tumor microenvironment. To counteract this phenomenon, lactate is re-taken up by the MCTs or the cellular buffering system is upregulated, such as carbonic anhydrases (CAs) [3,4]. The LDH enzyme is more efficient as an NAD^+^ regenerator when compared to respiratory chain complex I, partly explaining why glycolysis and fermentation are elevated in cancer cells. Indeed, NAD^+^(re)generation is necessary for the NAD^+^/NADH redox balance and for nucleotide and amino acid biosynthesis, thus generating a positive feedback loop and maintaining enhanced glycolytic activity in highly proliferative cells [5].

Pyruvate dehydrogenase kinases (PDHKs) are a family of mitochondrial enzymes which are in abundance in cells with high glycolytic activity, including most cancer cells, thus representing a therapeutic target [6]. PDHKs selectively inhibit pyruvate dehydrogenase complex (PDH), a key metabolic checkpoint between glycolytic and oxidative pathways, preventing pyruvate conversion into acetyl-coA, its entrance into mitochondria and thus redirecting pyruvate towards lactic fermentation. As a therapeutic strategy, disrupting pyruvate conversion to lactate by inhibiting PDHKs has been tested both in vitro for several cancers including non-small cell lung cancer model and glioblastoma (GB) differentiated cells (U87, U251, GL261) [7,8,9], as well as in preclinical models and clinical trials, with some promising results. The xenobiotic pyruvate analog dichloroacetate (DCA) has been broadly described as a PDHK inhibitor [10,11,12]. Safety-approved in patients from clinical trials, DCA is currently used in the treatment of acute and chronic lactic acidosis, inborn errors of mitochondrial metabolism, and diabetes [13].

In the present study, we showed that PDHK1 expression, and not PDHK2, was increased in high-grade glioma (HGG) tissues when compared with control or low-grade glioma (LGG) tissues, in two brain cancer cohorts. PDHK1 was also correlated to increased levels of glycolysis genes. PDHK1 and PDHK2 were found to be differentially expressed in specific GB areas, PDHK1 in hypoxic areas and PDHK2 in the invasive margin. Glycolytic activity was partially impaired in hypoxic-PDHK2 knockout (KO) and DCA-treated cells, although only DCA improved maximal cell respiration. We found that PDHK1 and PDHK2 KO impacted spheroid growth. PDHK2 KO additionally reduced cell invasion from spheroids. DCA also massively reduced spheroid growth and invasion, but surprisingly shortened mouse survival in an orthotopic GB xenograft model. Finally, increased mouse survival was observed when PDHK1 or PDHK2 KO GB stem-like cells were implanted, and cranial irradiation further improved mouse survival.

## 2. Materials and Methods

### 2.1. Ethical Issues

Male RAGγ2C^−/−^ mice were housed and treated in the animal facility of Bordeaux University. All animal procedures were performed according to the institutional guidelines and approved by the local ethics committee (agreement number: A5522). Patients gave their consent before tissue analysis according to the clinical guidelines. Informed written consent was obtained from all subjects at the Haukeland Hospital, Bergen, Norway.

### 2.2. Expression and Correlation Analysis

Correlations between gene expression levels of PDHK1 and PDHK2 and genes involved in glucose metabolism were analyzed in RNA sequencing cohort database from glioma tissue obtained from the cancer genome association (TCGA, GBMLGG cohort, *n* = 696), normal brain tissue obtained from TCGA (*n* = 5) and RNA sequencing data obtained from the Ivy Glioblastoma Atlas Project (IVYGAP) cohort consisting of high-grade glioma samples (*n* = 122) [14]. Specific areas of HGG tumors were distinguished based on IVYGAP microdissection analysis: cellular tumor corresponding to the tumor mass (ratio of tumor cells to normal cells = 100–500/1); infiltrating tumor representing the intermediate zone between the outermost boundary of the tumor and the cellular tumor (ratio of tumor cells to normal cells about 10–20/100); microvascular proliferation for the area surrounding two or more interconnected blood vessels (generally found in the core of tumors); and pseudopalisading cells around necrosis for a specific zone surrounding tumor core necrosis (tumor cells lining up in rows 10–30 nuclei wide at a higher density than cellular tumor). The expression level data are presented as FKPM log2 values. For the correlation analysis, values are displayed as Spearman’s correlation coefficient test Rho values.

### 2.3. Cell Culture

P3 cells were previously generated from the low passage of patient-derived xenografts and were stabilized as a stem-like cell line (characterized by IDH1 wt and MGMT unmethylated promoter [15]). Patient-derived glioblastoma cells P3 were cultured in neurobasal medium (NBM) supplemented with B-27 serum-free supplement (ThermoFisher Scientific), heparin (100 U/µL), penicillin/streptomycin (1000 U/mL) and basic FGF-2 20 ng/mL, in a 37 °C and 5% CO_2_ incubator. Parental cells were defined as P3 wt in the text. Stable cell lines were generated by lentiviral infection. To generate lentiviral particles, HEK293T cells were transfected with the lentiCRISPR v2 vector with psPAX2 (packaging construct), pMD2.G (viral envelope) and sgRNA targeting either PDHK1 or PDHK2. Empty lentiviral plasmids were used as controls for transducing and generating P3 sgCont cells. PDHK1 (fwd. 5′CACCGGTCATTCCCACAATGGCCCA 3′, rev 3′ CCAGTAAGGGTGTTACCGGGTCAAA 5′). PDHK2 fwd.5´ CACCGGCAGTTTCTGGACTTCGGTA 3′, rev 3′ CCGTCAAAGACCTGAAGCCATCAAA 5′). Culture medium was replaced by Opti-MEM with 20 mM HEPES 16 h post-transfection. Two to three days afterwards, the supernatant was collected, filtered (0.22 µm), supplemented with 8 µg/mL polybrene and finally used to infect P3 cells. The infected cells were selected using blasticidin (10 µg/mL). PDHK1 or PDHK2 deletion was further analyzed by Western blot. To evaluate the impact of different oxygen concentrations, cells were incubated either at 21% O_2_ (referred to as “normoxic conditions”) or 0.1% O_2_ to mimic hypoxia (hypoxia incubator Xvivo System model X1; BioSpherix Ltd., Parish, NY, USA). When specified, DCA treatment (Sigma-Aldrich, Saint-Quentin-Fallavier, France) was performed on our cells at a final concentration of 25 mM (the dose was selected based on the literature [7], and previous dose–response analysis on our P3 cell model).

### 2.4. Western Blotting

Cells were counted, seeded in a 6-well plate and exposed to different experimental conditions (treatments and/or different oxygen levels) for 48 to 72 h [16,17]. Cells were then washed with PBS and lysed in RIPA buffer (NaCl 150 mM, tris-base 50 mM, NP40 1% *w*/*v*, SDS 0.1% *w*/*v*, sodium deoxycholate 0.5% *w*/*v* and EDTA 1 mM pH 8) containing proteases and phosphatases inhibitors. Protein concentrations were determined using the BCA Protein assay kit (ThermoFisher Scientific, Waltham, MA, USA). Cell lysates were mixed with Laemmli buffer (60 mM Tris-HCl pH 6.8, 2% SDS *w*/*v*, 4% glycerol *w*/*v*, 9% thioglycerol *w/v*, 5% bromophenol blue *w*/*v)* and loaded on bis-tris acrylamide gels, before being transferred on a nitrocellulose membrane or PVDF (Amersham Biosciences GE Healthcare, Chicago, IL, USA). To perform immunodetection, membranes were blocked with 5% (*w*/*v*) milk diluted in Tris Buffer Saline with Tween 0.05% (TBS-T; pH 8), incubated with primary antibodies anti-PDHK1 (ab207450; Abcam, Cambridge, UK), anti-PDHK2 (PA5-35376; Invitrogen, ThermoFisher Scientific, Waltham, MA, USA), phospho-PDH (ab92696; Abcam, Cambridge, UK), anti-vinculin (V9131; Sigma-Aldrich, Saint-Quentin-Fallavier, France) and HRP-coupled anti-mouse/rabbit secondary antibodies (Jackson ImmunoResearch Europe Ltd., Ely, UK) were used. Immunodetection was performed using Clarity Western ECL Substrate (Bio-Rad, Hercules, CA, USA) and Amersham ImageQuant 800 (GE Healthcare, Chicago, IL, USA). Fiji software was used to determine and analyze densitometric profiles (version 1.53q; US National Health Institute, USA [18]). Full scans are included as Appendix A. Each Western blot experiment was repeated at least 3 times under the same conditions in order to provide a representative protein expression profile. Original Western Blot figures shown in Appendix A.

### 2.5. Immunofluorescence Experiments

To plate P3 cells for immunolabeling, coverslips were first coated with Matrigel (Corning^TM^ Matrigel^TM^ growth factor reduced membrane matrix, 356231; ThermoFisher Scientific, Waltham, MA, USA) at a concentration of 0.2 mg/mL. P3 SgCont, sgPDHK1 or sgPDHK2 cells were plated on the coated coverslips and incubated for 24 to 72 h, depending on the conditions tested (25 mM DCA treatment and/or hypoxia). Cells were then washed with phosphate-buffered saline (PBS), fixed with 4% paraformaldehyde (PFA) for 30 min and washed again with PBS. Fixed cells were permeabilized with 0.1% Triton x-100 (diluted in PBS) for 5 min and washed 3 times with PBS. After 10 min incubation in blocking solution (1% bovine serum albumin and 2% fetal bovine serum in PBS), samples were incubated with primary antibodies (anti-Tom20, sc-17764; SantaCruz Biotechnology, Dallas, TX, USA) in blocking buffer for 1 h, washed 3 times with PBS and then incubated for 30 min with blocking buffer supplemented with appropriate secondary fluorescent antibodies (Goat anti-Rabbit IgG (H + L,) Alexa Fluor Plus 555, 15636746, and Goat anti-Mouse IgG (H + L) Alexa Fluor Plus 488, 15626746; ThermoFisher Scientific, Waltham, MA, USA), Hoechst 33342 (H3570; ThermoFisher Scientific, Waltham, MA, USA), and/or phalloidin rhodamine (R415; ThermoFisher Scientific, Waltham, MA, USA). After PBS washes, samples were mounted using Prolong Gold antifade reagent (P36930; ThermoFisher Scientific, Waltham, MA, USA). Image processing of cells and mitochondria was performed with a homemade macro in Fiji software, either from [17] or adapted from [19].

### 2.6. Seahorse Oxygen Consumption and Extracellular Acidification Measurements

The Seahorse XFe96 Analyzer (Agilent, Santa Clara, CA, USA) was used to measure the oxygen consumption rate (OCR) and extracellular acidification rate (ECAR). The cell number and concentration of CCCP concentrations were titrated before running the assay. All compounds were obtained from Sigma-Aldrich, France, if not otherwise stated. The mitostress assay medium was D5030 supplemented with 10 mM glucose, 2 mM pyruvate and 4 mM glutamine at pH 7.4 (+/− 0.05). Throughout the mitostress experiments, we used 3 µM oligomycin (added in port A), 2.5 µM CCCP (added in port B), 1 µM rotenone (added in port C) and 1 µM antimycin A (added in port D). Data were ROX-corrected (antimycin A, non-mitochondrial respiration). Upon investigating the acute effects of DCA, 20,000 cells were plated the day before the assay in Seahorse 96-well plates. Mitostress assay medium was used during the assay. The pH of the DCA addition was carefully adjusted before performing the assay. For the real-time measurement of OCR and ECAR, ddH20 control or 20 mM DCA was added in port A.

### 2.7. Oroboros Oxygen Consumption Assay

Real-time oxygen consumption of intact P3 cells (SgCont, SgPDHK1 or SgPDHK2, +/− DCA 25 mM) was also measured with an Oxygraph-2 k (Oroboros Instruments, Innsbruck, Austria). When indicated, cells were incubated with DCA at the final concentration of 25 mM for 3 days. Oxygen consumption was measured at 37 °C in cell culture medium. The oxygen consumption rate was measured under 3 different conditions: phosphorylating state (endogenous respiratory condition), non-phosphorylating state with addition of oligomycin (50 ng/mL), and uncoupled state by the successive addition of carbonyl cyanide m-chlorophenyl hydrazone (CCCP 0.5 μM) to reach the maximal respiration. The determine the sensibility to complex I and complex II, injections of rotenone (30 nM) or atpenin A5 (AtpnA5, 20 nM) were performed, respectively [20,21]. Antimycin A (100 nM) was added at the end of each experiment to evaluate the proportion of oxygen consumption not linked to the respiratory chain.

### 2.8. ROS Detection Method

CellROX Oxidative Stress Reagents (CellROX Green Reagent, C10444; Life technologies, Carlsbad, CA, USA) were used to detect reactive oxygen species (ROS) on P3 spheroids. Cells (SgCont, SgPDHK1 or SgPDHK2) were first incubated for 3 days, with or without DCA treatment (25 mM). On the third day, CellROX probe was added directly to the cell medium and incubated for 30 min. Spheroids were then washed with PBS, dissociated with accutase (5 min) and resuspended again in PBS. Green fluorescence of CellROX probe, proportional to ROS production, was assessed for each condition on 10,000 events by flow cytometry (BD Accuri™ C6 Flow Cytometer; BD Biosciences, Franklin Lakes, USA). Positive control (1,1-dimethylethyl hydroperoxide; CAS 75-91-2, Sigma-Aldrich, Saint-Quentin-Fallavier, France) was also added as an internal control for each replicate (data not shown). Analysis of flow cytometry data was performed with FlowJo software (v10.8.1; Ashland, OR, USA).

### 2.9. Lactate Measurements

Lactate measurement was performed as previously described [22]. In brief, P3 cells (0.5 to 10^6^ cells) were seeded in a 6-well plate with different conditions: SgCont, SgPDHK1, SgPDHK2, with or without DCA treatment (25 mM) in a final volume of 2 mL. Cells were incubated for 72 h either in a 21% or 0.1% O_2_ incubator (hypoxia incubator Xvivo System model X1; BioSpherix Ltd., Parish, NY, USA). The day of the experiment, cell pellets and cellular medium were separated by centrifugation. Cellular pellets and medium were collected and separated by centrifugation. Pellets and medium were resuspended with perchloric acid 14% (PCA w/o EDTA; Merck Millipore, Burlington, MA, USA). Pellets and medium were neutralized with KOMO solution (KOH 2 M, MOPS 0.5 mM in H_2_O; VWR, Radnor, PA, USA). Lactate enzymatic measurement was then performed in every sample using a mix of L-Lactate oxidase enzyme (0.5 U/mL; Merck Millipore, Burlington, MA, USA), peroxidase from horseradish enzyme (HRP) 5 U/mL (77332; Sigma-Aldrich, Saint-Quentin-Fallavier, France) and Amplex Red reagent 2.5 µM (A12222; ThermoFisher Scientific, Waltham, MA, USA) in a Tris-HCl buffer (50 mM in H_2_O, pH 6.8 to 7).

### 2.10. Spheroid Experimental Assays

All the protocols used for spheroid assays were previously described by Guyon et al. [23]. Briefly, 10^4^ cells per well were seeded in cell medium supplemented with 0.4% methylcellulose in a 96-well round-bottomed plate to obtain a single spheroid after 3 days of incubation at 37 °C, 5% CO_2._ For the spheroid growth assay, cell medium supplemented with DCA or vehicle was added on the top of each spheroid after the 3 days of formation. Spheroid growth was monitored for 7 days at 21% O_2_ or 0.1% O_2_, every 24 h or 48 h, by brightfield image acquisition with a microscope (Olympus IX81). For the spheroid invasion assay, each spheroid was washed in PBS and included in a 96-well flat-bottom plate, in 100 µL of cold type I collagen matrix (1 mg/mL collagen type I diluted in PBS and NaOH). After complete polymerization (30 min at 37 °C), cell medium containing DCA or vehicle was added. Spheroid invasion was assessed after a 24 h incubation at 21% or 0.1% O_2_.

### 2.11. Intracranial Implantation

To perform in vivo experiments, 5 spheroids of knockout P3 cells for PDHK1 or PDHK2 and control cell lines were stereotaxically implanted into the striatum of immunodeficient RAGγ2C^−/−^ mice. Mice were housed and treated in the animal facility of Bordeaux University (“Animalerie Mutualisée Bordeaux”). The animals were anesthetized with ketamine (1.5 mg/kg) and xylazine (150 μg/kg), a burr hole was drilled 2.2 mm to the left of the bregma and the spheroids were implanted into the dorsal striatum at 3 mm depth using a 10 μL Hamilton syringe. An analgesic procedure was applied with the subcutaneous injection of buprenorphine (0.1 mg/kg, 10 min before and once 12 to 24 h after implantation). On day 4, the injection protocols started: P3 SgCont implanted mice were treated daily either with DCA (25 mg/kg in saline solution, ip) or saline solution (vehicle). Where indicated, animals were subjected to a radiotherapeutic protocol (3 × 2 Gray). At the end of the experiment, animals were euthanized by cervical dislocation and brains removed for immunohistological analysis.

### 2.12. Immunohistological Procedures

Brains were extracted, immediately frozen in liquid nitrogen and stored at −80 °C. Sections (10 µm) of cryopreserved brains were prepared using a cryostat (Leica, Wetzler, Germany) and mounted on slides. For the immunolabeling processes, cryo-sections were dried at room temperature for 10 min and then fixed with 4% PFA for 15 min. Slides were washed 3 times with PBS and permeabilized with 0.1% Triton x-100 (15 min). After 3 washes with PBS, samples were blocked with 1% bovine serum albumin and 2% fetal bovine serum in PBS (blocking solution) for 30 min. Samples were then incubated overnight (4 °C) with primary antibodies in blocking solution, followed by 3 washes with PBS and incubation in secondary antibodies in blocking solution for 1 h 30. Human anti-nestin (MA1-110; Invitrogen, ThermoFisher Scientific, Waltham, MA, USA) primary antibody was used to detect patient-derived tumor cells and Hoechst and/or rhodamine phalloidin (Invitrogen, ThermoFisher Scientific, Waltham, MA, USA) staining to identify nucleus and/or cytoskeleton. Samples were washed 3 times with PBS and mounted using Prolong Gold antifade reagent. The slide scanner was a Nanozoomer 2.0HT with a fluorescence imaging module (Hamamatsu Photonics, Massy, France) using objective UPS APO 20X NA 0.75 combined to an additional lens 1.75X, leading to a final magnification of 35X. Virtual slides were acquired with a TDI-3CCD camera. Fluorescent acquisitions were performed with a mercury lamp (LX2000 200W—Hamamatsu Photonics, Massy, France) and the set of filters adapted for DAPI, and/or GFP/Alexa 488, and/or Alexa 568 and or Alexa 647/Cy5 fluorescence. Image analysis was performed with NDP.view2 Viewing software (Hamamatsu Photonics, Massy, France).

### 2.13. Statistics and Figures

Data were analyzed using GraphPad Prism 8 software and figures were prepared with Inkscape software. Depending on the conditions tested, one- or two-way ANOVA and unpaired two-tailed Student’s *t*-tests were used to evaluate statistical differences between the samples. *p*-values < 0.05 were considered to be statistically significant. For correlation analysis, values are displayed as Spearman’s correlation coefficient test and rho values were considered significant when >0.3 and <−0.3.

## 3. Results

### 3.1. PDHK1 Is Increased in Glioblastoma Patient Samples and Correlates with Glucose Metabolism and pH Regulation Genes

Transcriptional analysis from TCGA patient cohorts revealed that the expression of PDHK1 was significantly higher in GB samples than in low-grade glioma (LGG) samples (Figure 1A). Distribution analysis of PDHK1 expression extracted from IVYGAP dataset revealed a high expression in pseudopalisading cells around necrosis, known as the hypoxic area (Figure 1B). In contrast, PDHK2 expression was lower in GB than in LGG (Figure 1C). Distribution analysis of PDHK2 expression showed that infiltrating tumor areas were enriched in PDHK2, when compared with GB cellular tumor (Figure 1D). Correlation analysis of the dataset from TCGA, as well as from the IVYGAP dataset, revealed that PDHK1 expression was correlated with several genes involved in glycolysis and glucose metabolism, including the glucose transporters (GLUT1 and 2), LDHA and the lactate transporter MCT1 (SLC16A1) (Figure 1E). In contrast, PDHK2 expression was correlated with PFKM or PFKFB2 in both datasets (Figure 1E). In the LGG dataset, there was no correlation with the vast majority of the genes involved in glucose metabolism (Figure 1E). For two key genes involved in pH regulation, carbonic anhydrase IX and XII (CAIX and CAXII), a strong and significant correlation was found between PDHK1 for CAIX (Figure 1F) and CAXII (Figure 1G), but not for PDHK2, in both datasets in GB tissues, indicating potential different roles of PDHK1 and PDHK2 (Figure 1F,G). CAIX and CAXII expression was significantly correlated with the glucose metabolism genes (Figure 1H) and was also higher in pseudopalisading cells around necrosis (Figure 1I) as for GLUT1, LDHA, and SLC16A1 (MCT1) expression (Figure 1K–M). Taken together, these results showed that PDHK1 is elevated in GB hypoxic areas, whereas PDHK2 is higher at the invasive front.

### 3.2. Inhibition of PDHK Activity by DCA Modulates PDHK Expression and Cell Morphology

To determine the role of PDHK1, PDHK2 and PDH phosphorylation in GB development (Figure 2A), we used a stem-like cell model (P3 cells), as previously characterized [16,24]. P3 cells were transduced with CRISPR-Cas9 constructs against either *PDHK1* or *PDHK2* gene, and protein depletion was confirmed by Western blot analysis at 21% and 0.1% O_2_ (Figure 2B). PDHK1 expression, but not PDHK2, was found to be increased after hypoxia incubation (Figure 2B,D). Furthermore, PDH phosphorylation status was analyzed in PDHK1 and PDHK2 knockout cells and was found to be decreased compared with P3 SgCont cells in hypoxic conditions (Figure 2C). Then, P3 cells were treated with 25 mM of dichloroacetate, (a large-range PDHK inhibitor), for 72 h, and cultured either at 21% or 0.1% O_2_. Upon DCA treatment, PDHK1 expression remained stable at 21% O_2_ but decreased in hypoxic conditions (Figure 2D). PDHK2 levels were significantly and equally decreased upon DCA treatment in both oxygen conditions (Figure 2D). At 0.1% O_2_, PDH phosphorylation was increased in P3 wt cells and inhibited upon DCA treatment (Figure 2E).

Cell morphology analysis showed that PDHK2 KO and DCA-treated cells increased in size. DCA-treated cells were also found to be more elongated, as shown by an increase in aspect ratio and form factor parameters (Figure 2F–H), correlating with a higher number of microtubes (Figure 2F, lower panels), membrane structures that have been described previously [24]. Finally, the mitochondrial network was analyzed based on anti-Tom20 staining and no difference was observed in the mitochondrial mass between conditions, but the mitochondrial networks appeared more fragmented in PDHK2 KO and DCA-treated cells (Figure 2I), indicating mitochondrial dysfunction.

### 3.3. Inhibition of PDHK Activity Impacts GB Cell Energy Metabolism

To investigate the metabolic effect of both DCA and single PDHK knockout in the GB cells, respiration capacity and lactate production assays were performed. The acute metabolic effect of 25 mM DCA on P3 cells showed that the extracellular acidification rate (ECAR), predominantly reflecting extracellular lactate, was significantly reduced (Figure 3A). To confirm this result, specific lactate measurements were performed on P3 cells, treated with 25 mM DCA or vehicle for 72 h. Both cell pellets and supernatants were collected to enable measurements of intracellular production and extracellular secretion of lactate, respectively. Intracellular lactate pools were significantly increased at 0.1% O_2_ compared with 21% O_2_, both for P3 control and PDHK1 KO cells. DCA reduced the production of intracellular lactate in both oxygen conditions; however, this was only significant at 0.1% O_2_ (Figure 3B), as also observed for PDHK2 KO cells. Consistently, the level of extracellular lactate increased in P3 control cells upon incubation at 0.1% O_2_, as well as in P3 PDHK1 and PDHK2 KO cells, and lactate secretion was significantly reduced upon DCA treatment (Figure 3B).

Furthermore, acute DCA treatment significantly increased the oxygen consumption rate (OCR) for the P3 cells (Figure 3C). Additional oxygen consumption analysis was performed using the Oroboros oxygraph methodology and showed no difference in endogenous oxygen consumption rate between all conditions, even with a long-term DCA treatment (3 days) (Figure 3D). However, the maximal respiratory capacity (uncoupled conditions) was significantly increased upon DCA treatment, but not altered in PDHK1 or PDHK2 KO cells, compared with control cells (Figure 3D). Moreover, DCA-treated cells were detected as more sensitive than non-treated cells to rotenone, an inhibitor of oxidative phosphorylation (OxPhos) complex I (Figure 3E), indicating that DCA induced electron transport chain (ETC) activity more dependent on complex I (Figure 3E). PDHK1 or PDHK2 KO cells did not present differences in respiration chain inhibitor sensitivity (Figure 3E), suggesting a compensatory effect between PDHKs on cell respiration. ROS production was then measured using the CellROX probe and a significant increase was detected in DCA-treated cells, but not in PDHK1 or PDHK2 KO cells (Figure 3F).

### 3.4. Disruption of PDHK1 or PDHK2 Activity Impairs GB Proliferation and Invasion

P3 control cells treated with DCA or vehicle and PDHK1/PDHK2 KO cells were tracked for spheroid growth during a period of 7 days, reflecting cell proliferation. The growth rate was higher in spheroids incubated at 21% O_2_ when compared with 0.1% O_2_ (Figure 4A). PDHK1 and PDHK2 KO spheroids presented a lower growth rate than the corresponding control spheroids at both 21% and 0.1% O_2_ (Figure 4A). DCA drastically impaired spheroid growth (Figure 4A) and destabilized spheroid integrity at 0.1% O_2_, as observed by image monitoring and by viability assay (Figure 4B, right panels). In contrast, control spheroids presented a compact shape with few detached cells (Figure 4B, left panels). We then performed invasion experiments by embedding spheroids in collagen I gel (Figure 4C). Both PDHK1 KO, PDHK2 KO and DCA-treated spheroids showed a decreased invasion potential as compared with SgCont spheroids at 21% O_2_ (Figure 4C, right upper panel). At 0.1% O_2_, invasion potential was decreased by almost 50% after DCA treatment, and by almost 25% for PDHK2 KO spheroids, when compared with the control (Figure 4C, right lower panel). Furthermore, DCA-treated cells presented long and thin protrusions around the spheroid core, resembling 3D microtube structures (Figure 3C).

### 3.5. Simple Knockout of PDHK1 or PDHK2 Impairs In Vivo GB Tumor Growth and Invasion and Improves Mice Survival

To evaluate the in vivo effects of PDHK modulation on tumor growth, intracranial implantations of control, PDHK1 KO or PDHK2 KO P3 spheroids were performed in immunodeficient mouse brains. A group of control-implanted mice was treated with 25 mg/kg DCA daily, whereas the others were injected with the vehicle. Histological analysis of the tumors from the different groups showed a decrease in tumor core area in PDHK1 and PDHK2 KO groups (Figure 5A, upper panel), and the invasive area was smaller only in PDHK2 group (Figure 5A, lower panel). No detectable difference in tumor core or invasive area was observed in the DCA-treated group when compared with the control condition (Figure 5A). Contralateral hemisphere invasion was not detected in the PDHK2 KO group, in contrast to the other groups (Figure 5B). Surprisingly, mice treated with DCA exhibited shorter survival than the control group, but both PDHK1 and PDHK2 KO induced an improvement in overall survival, with a gain of approximately 10 days when compared with the control group (Figure 5C).

Finally, in order to compare to GB treatment in patients, a radiotherapeutic protocol was used in an in vivo experiment to evaluate a potential additive effect of irradiation on the PDHK KO GB model. Comparable survival curves were observed between PDHK1 and PDHK2 KO groups, and PDHK1 is preferentially expressed in tumor cores, which is specifically targeted by cranial irradiation; therefore, mice from control and PDHK1 groups were selected for a three-time 2 Gy irradiation protocol. Histological analysis of core and invasive areas showed that irradiation decreased the tumor mass area and invasive capacities (Figure 5D). Irradiated PDHK1 KO tumors had even smaller central and invasive areas than the control tumors (Figure 5D). A similar trend was observed in the survival curves, with irradiation improving mouse survival in the two groups (SgCont and SgPDHK1) without inducing a synergistic effect (Figure 5E).

## 4. Discussion

Adapting metabolism according to the needs of cancer cells is crucial for tumor growth. This process, defined as the “metabolic reprogramming” hallmark of cancer [1], aims to sustain intracellular homeostasis and increase biomass and energy availability.

PDHKs specifically regulate the activation of the PDH complex, which controls the conversion of pyruvate into acetyl-coA to fuel the Krebs cycle, and consequently, the electron transport chain in mitochondria. PDHK activation impacts the balance between metabolic fluxes and promotes glycolytic pathways [25]. Indeed, PDHKs are found to be dysregulated in several cancer cell types including gliomas, as recently reviewed by Anwar et al. [26]. Here, we further investigated PDHK expression in patient samples from different glioma grades and found that the expression of PDHK1 was elevated in high-grade gliomas compared to low-grade gliomas. PDHK2 however, was found to be highly expressed in most of glioma samples. Moreover, the IVYGAP dataset on GB samples showed a differential intratumoral distribution of PDHK1 and PDHK2, suggesting distinct roles for the two isoforms. PDHK2 was preferentially expressed in the invasive area of the tumor, in contrast to PDHK1, finely regulated by HIF1α, and which was more expressed in pseudopalisading cells surrounding necrotic areas. We also found a positive correlation between PDHK1 expression and genes involved in glucose transport (GLUT-1) and metabolism (several enzymes of the glycolysis such as ALDOA), as well as in lactate production (LDHA) and secretion (SLC16A1/MCT1). Key regulators of the pH homeostasis system, such as CAIX and CAXII, were also correlated with the expression of PDHK1. Together, these findings indicate that high glycolytic flux observed in cancer cells, increasing lactic acid production and deacidification mechanisms to avoid intracellular acidosis [4], is correlated to PDHK1 expression, confirming its role in sustaining the glycolysis-dependent proliferation of cancer cells. Thus, PDHK1, expressed in the tumor core, could be a potential candidate in controlling the balance between oxidative phosphorylation and lactic fermentation, and PDHK2 in targeting invasion.

To study the role of PDHKs in GB, we used patient-derived stem-like cells (P3 cells), in which we confirmed that hypoxic conditions (0.1% O_2_) regulate PDHK1 but not PDHK2 protein expression. We showed that the pharmacological inhibition of PDHKs by DCA reduced the expression of both isoforms, delineating a potential effect of DCA on PDHK protein stability. Phosphorylation of the E1α subunit of PDH residue S293, reflecting PDHKs activity, was increased in hypoxia and was inhibited by DCA, as previously described [27]. To dissect the differential role of PDHK1 and PDHK2, we developed PDHK single knockout cells using CRISPR-Cas9 technology. At 0.1% O_2_, PDHK1 and PDHK2 KO cells showed a lower level of PDH phosphorylation when compared with the control condition. Interestingly, at 21% O_2_, PDHK1 KO cells had higher p-PDH level than control cells, suggesting a compensatory effect of other PDHKs isoforms, potentially PDHK2. These results support our hypothesis of a specific PDHK isoform intratumoral organization, between hypoxic and more oxygenated tumor areas. Precise mechanisms remain to be elucidated.

Inhibition of PDHKs, either genetically or by using DCA treatment, impacted the extracellular acidification rate, as also shown in other models [28,29]. We confirmed that DCA reduced lactic acid production and secretion in our model related to increased glycolytic flux (hypoxic conditions). Oxidative pathways were also modulated; the oxygen consumption rate (OCR) was found to be increased after acute DCA treatment. Neither PDHK1 nor PDHK2 KO modulated the OCR; however, in future studies, investigating whether PDHK1/PDHK2 double KO modifies the OCR would be of interest. In contrast, basal OCR upon longer-term DCA treatment was not modified but was elevated under uncoupled conditions, suggesting a downstream control of ATP synthase on basal respiratory rate. Moreover, DCA-treated cells showed an increased sensitivity to rotenone, technically assessed as previously described [20], revealing the pivotal role of mitochondrial complex-I-dependent respiration. The resulting flux reorganization towards complex I led to increased ROS production in DCA-treated cells [30,31]. The metabolic rearrangement induced by PDHK inhibition also modified the P3 cell shape and mitochondrial network. This network appeared fragmented upon DCA treatment and in PDHK2 KO cells, a phenomenon observed when cells are exposed to important metabolic stress [32]. Cells treated with DCA also showed an elongated shape, losing cell–cell interaction when exposed to hypoxia. However, viability assays performed on spheroids treated with DCA for 7 days and incubated at 0.1% O_2_ did not show increased cell death as compared with vehicle-treated spheroids, suggesting a cytostatic but not cytotoxic effect of DCA, correlating with previously published data [12]. It has been reported that combining the inhibition of HIF1α with chrysin, and blockading PDHK activity with DCA, impairs in vitro GB growth in hypoxia [33]. Consistently, we showed that the pharmacological and genetic disruption of PDHKs strongly impaired GB spheroid growth. Notably, at both O_2_ levels, PDHK2 knockout and DCA treatment lowered the invasion rate when compared with the control condition, further supporting the role of PDHK2 in the invasive process.

DCA has been proposed in several pharmacological protocols, and results from clinical phase I trial confirmed the good tolerance of orally administered DCA to patients with recurrent brain tumors [34]. A Phase IIA trial of oral DCA will be performed in 40 patients before debulking surgery to evaluate the impact of DCA on recurrent GBs (https://clinicaltrials.gov/ct2/show/NCT05120284 (accessed on 24 July 2022)). However, the scientific community still seems to be cautious about the use of a global PDHK inhibitor [35]. Pre-clinical studies emphasized DCA side effects, possibly linked to the hyperactivation of oxidative pathways and protumoral outcomes [36,37]. In our study, we evaluated the impact of DCA monotherapy treatment on mice implanted with P3 stem-like cells in comparison with a single knockout of either PDHK1 or PDHK2. We found that the tumor area in the DCA-treated group was not significantly different when compared with the control group. We hypothesize that differences observed between in vitro and in vivo conditions for the DCA treatment effectiveness may be linked to higher in vivo antioxidant defenses, provided either by GB stem-like cells themselves or by the cerebral tumor microenvironment. Furthermore, the stem-like cell model used in this study exhibited greater metabolic adaptation capacities [38,39] than other GB cell line models, such as U87 or U251, which are differentiated (serum-cultured cells) and mainly glycolytic cells. Consistently, a metabolic switch towards mitochondrial respiration induced by DCA is more effective in reducing tumor progression in these glycolytic models when compared with the P3 stem-like cells used in our study. To reinforce the in vivo results obtained in P3 tumors after DCA treatment, future studies on other patient-derived stem-like cell lines will be conducted.

Moreover, even when using a lower DCA dose than previously reported in vivo studies [40,41], the survival of DCA-treated mice was shortened in comparison to the control group, as also shown in other glioblastoma models (e.g., GL261) [9]. These results highlight a possible undesirable and undetermined adverse effect of DCA in our model and emphasize the need for a novel therapeutic strategy. An alternative approach would be to combine DCA with radiotherapy or concomitant medication that could potentiate the anti-cancer properties of DCA, and thus enhance the benefit/risk ratio. Indeed, DCA has already been shown to sensitize U87 cells to radiotherapy by reducing the mitochondrial reserve capacity and increasing oxidative stress [40]. In addition, considering the potential toxic effect of DCA shown in our model, PDHK-isoform-specific inhibition could be a more promising and safer therapy. The simple knockout of PDHK1 or PDHK2 clearly reduced GB progression in vitro, but also in vivo tumor growth when compared with the control group; irradiation amplified this phenomenon without synergizing it. Interestingly, PDHK2 KO tumors did not present any contralateral hemisphere invasion, reinforcing a potential role of PDHK2 in GB invasion, in line with reduced invasion capacities observed in vitro. Considering the central role of GB stem-like cells in invasion, treatment resistance and GB relapse [42,43], these therapeutic options would represent a benefit for GB patients. Targeting PDHKs remains promising and alternatives to DCA, such as isoform-specific inhibitors, DCA mimetics or drugs targeting other sites of the PDHKs, are being developed [44].

More recently, in the era of immunotherapy in cancer medicine, DCA treatment has also proved to enhance the anti-tumor immune response in hepatocellular carcinoma [45]. Reduced production and secretion of lactate resulted in deacidification of the microenvironment, allowing for an innate immune response, representing an additional antitumor effect of DCA [46]. Immunotherapy treatment, such as immune checkpoint inhibitor (ICI), has failed in GB patients [47], but combinatory treatment with DCA could represent an interesting therapy potentiating axis; other therapeutic combinations with ICIs are currently in clinical trials [48].

## 5. Conclusions

In the present study, we found PDHK1 and PDHK2 to have distinct expression patterns and functions in GB tissue, coherent with metabolic specific organization and heterogeneity well characterized in GB [49]. We specifically described a new role of PDHK2 in GB invasion and lactate production. We recently showed that lactate fuels the TCA cycle to improve invasive capacities [17]; therefore, it will be interesting to further study and decipher mechanisms linking lactate, PDHK2 activity and GB invasive capacities. In a pre-clinical orthotopic mouse model, we found that both tumors generated from PDHK1 KO and PDHK2 KO cells prolonged overall mouse survival. According to our results, future therapeutic strategies to treat GB could include specific inhibitors of PDHK1 and PDHK2, which could be used sequentially to target specific roles of each isoform in GB progression, and/or be combined with radiotherapy. Distinct roles of PDHK1 and PDHK2 in GB biology will be elucidated in future studies: as they are differentially expressed in GB tumors, their specific inhibition could probably destabilize metabolic organization between the tumor central area in which PDHK1 is more expressed and the invasive area enriched for PDHK2 expression.

DCA decreased GB proliferation and invasion in our in vitro GB model but did not prolong survival when used as monotherapy in vivo. This may be due to a high metabolic flexibility of GB stem-like cells or symbiotic effects from cells in the microenvironment. Further evaluation of DCA treatment in a preclinical radio/chemotherapeutic protocol would be of interest. Overall, our findings indicate that the manipulation of PDHKs in GB shows potential for novel treatment strategies.

## Figures and Tables

**Figure 1 cancers-14-03769-f001:**
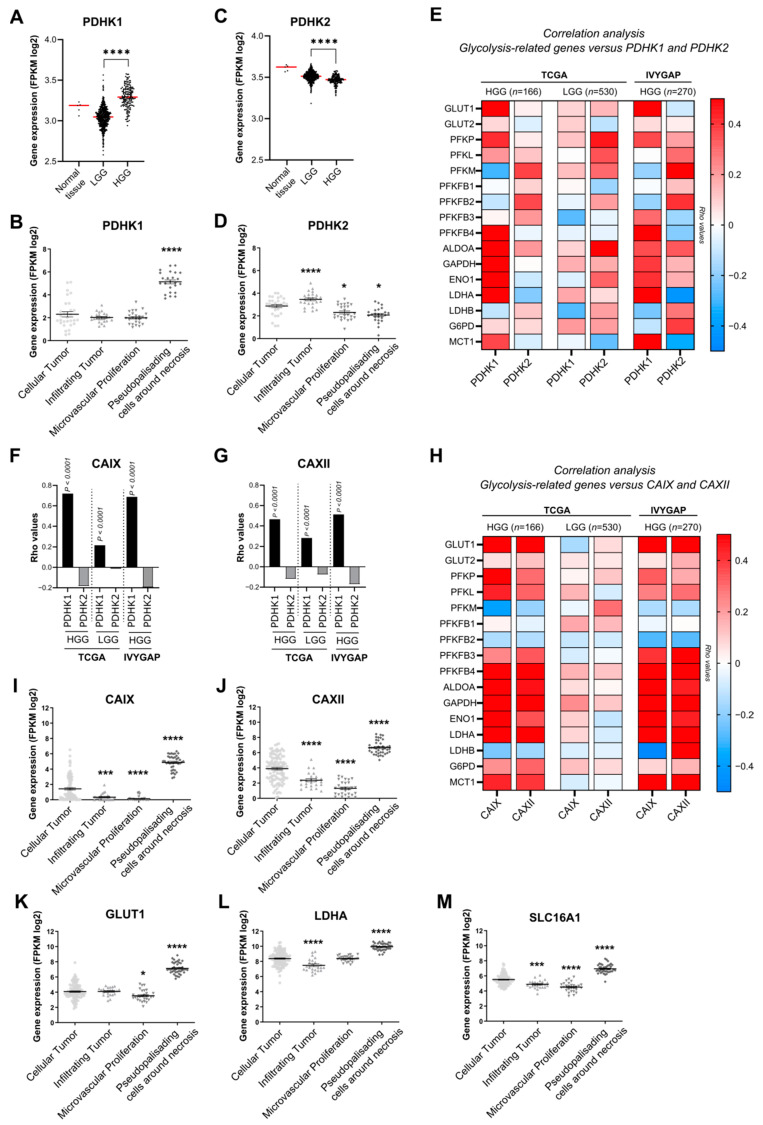
Expression and correlation analysis of PDHK1, PDHK2 and genes involved in glucose metabolism in HGG and LGG. (**A**) Expression levels of PDHK1 in normal tissue, low-grade glioma (LGG) and high-grade glioma (HGG) (TCGA). (**B**) PDHK1 expression levels and localization in HGG tissue (IVYGAP). (**C**) Expression levels of PDHK2 in normal tissue, LGG and HGG (TCGA). (**D**) PDHK2 expression levels and localization in HGG tissue (IVYGAP). (**E**) Correlation analysis of genes involved in glycolysis with PDHK1 and PDHK2 in HGG and LGG from TCGA and HGG from TCGA + IVYGAP. (**F**) Correlation analysis of CAIX with PDHK1 and PDHK2 in HGG and LGG from TCGA and HGG from IVYGAP. (**G**) Correlation analysis of CAXII with PDHK1 and PDHK2 in HGG and LGG from TCGA and HGG (TCGA + IVYGAP). (**H**) Correlation analysis of genes involved in glycolysis with CAIX and CAXII (TCGA + IVYGAP). (**I**) CAIX and (**J**) CAXII expression levels and localization in HGG tissue (IVYGAP). (**K**) GLUT1, (**L**) LDHA and (**M**) SLC16A1 expression levels and localization in HGG tissue defined by the IVYGAP database. Data are expressed as the mean ± SEM and *p*-values were obtained using unpaired *t*-tests (**A**,**B**), Spearman’s correlation coefficient tests (**E**–**H**) and one-way ANOVA (**B**,**D**,**I**–**M**); * *p* < 0.05; *** *p* < 0.001; **** *p* < 0.0001.

**Figure 2 cancers-14-03769-f002:**
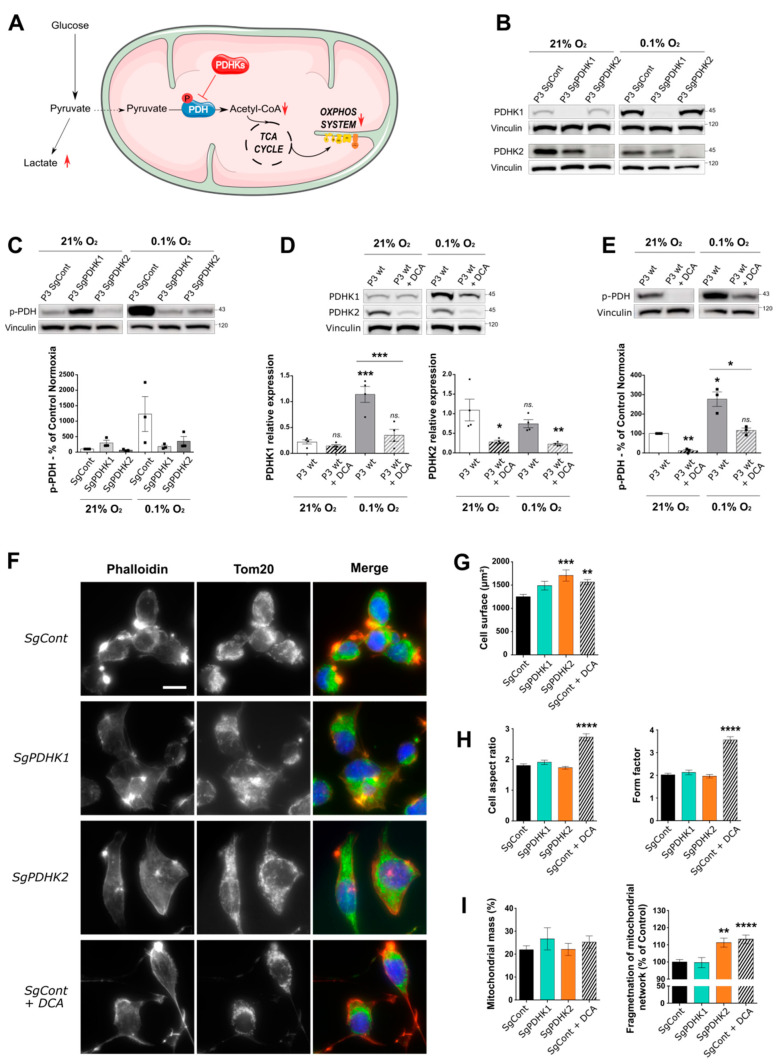
Dichloroacetate (DCA) reduces levels of phosphorylated PDH in a primary glioblastoma in vitro cell model, impacting cell and mitochondrial organization. (**A**) Illustration of the two alternative pyruvate utilization pathways and the impact of PDHK activity on the balance between lactic fermentation or mitochondrial oxidation of pyruvate. (**B**–**E**) Western blotting and corresponding quantification of PDHK1, PDHK2 expression or phospho-PDH status in P3 cells control, knockout for PDHK1, PDHK2 or treated with DCA 25 mM for 3 days; all cells were exposed to either 21% O_2_ or 0.1% O_2_. Graphs represent densitometry quantification normalized with vinculin (*n* = 3 or 4). (**F**) Epifluorescence images of immunofluorescence stainings on P3 SgCont, SgPDHK1, SgPDHK2 or treated with DCA 25 mM for 3 days (Phalloidin, red; Tom20, green; Hoechst, blue). Scale bar: 20 µm. (**G**,**H**) Immunofluorescence image analysis (120 to 240 cells per group): (**G**) cell surface quantification; (**H**, *left*) cell aspect ratio corresponding to the ratio of the major axis to minor axis of the cells; (**H**, *right*) form factor imaging branching of the cells. (**I**) Mitochondrial network analysis based on immunofluorescence pictures; (*left*) mitochondrial mass expressed as the percentage of total cell area (39 to 99 cells per group); (*right*) quantification of mitochondrial network fragmentation corresponding to the ratio of the number of mitochondria per cell on the area of the full mitochondrial network, normalized by the mean of control group and expressed as the percentage of control. Data are expressed as the mean ± SEM and *p*-values were obtained using one-way ANOVA (**D**,**G**,**H**,**I**
*left*) or unpaired *t*-tests (**C**,**E**,**I**
*right*); * *p* < 0.05; ** *p* < 0.01; *** *p* < 0.001; **** *p* < 0.0001; *ns.*: not significant.

**Figure 3 cancers-14-03769-f003:**
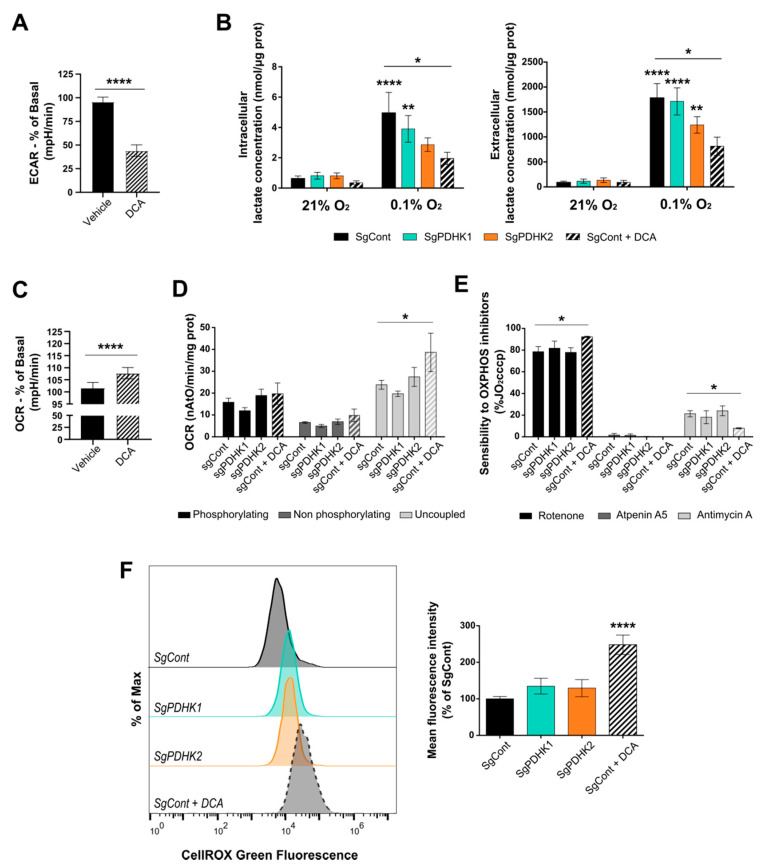
DCA disturbs GB cell metabolism and induces ROS production. (**A**) The extracellular acidification rate (ECAR) was measured in real time upon the addition of 20 mM DCA/vehicle to P3 wt cells. Values are presented as the percentage change compared with basal ECAR prior to DCA addition. (**B**) Intra- (*left*) and extracellular (*right*) lactate measurement in P3 SgCont, SgPDHK1, SgPDHK2 or with DCA treatment (25 mM for 3 days), upon 21% or 0.1% O_2_ exposure; *n* = 7 to 8 for each condition; when not noticed, significance was assessed by comparing with control cells 21% O_2_. (**C**) The oxygen consumption rate (OCR) was measured in real time upon the addition of 20 mM DCA/vehicle to P3 wt cells. Values are presented as the percentage change compared with basal OCR prior to DCA addition. (**D**) Oxygraphy analysis of P3 cells in a phosphorylating state (basal cell respiration), non-phosphorylating state (oligomycin addition) or uncoupled state (CCCP addition); *n* = 4. (**E**) Sensitivity in P3 cells to specific OxPhos complex inhibitors rotenone (complex I), atpeninA5 (complex II) and antimycin A (complex III); *n* = 4. The addition of inhibitors was performed upon uncoupled respiration conditions and results are expressed as the percentage of uncoupled respiration prior inhibitor addition (%JO2cccp). (**F**) ROS production assessed in triplicates with CellROX Green probe in PDHK1 or PDHK2 KO cells, or in control cells exposed to 3 days of DCA treatment (25 mM). *Left panel*: Representative histograms in modal scaling (percentage of Max). *Right panel*: Quantification of mean fluorescence intensity, expressed as the percentage of control cells. Data are expressed as the mean ± SEM, and *p*-values were obtained using unpaired *t*-tests (**A**,**C**), one-way ANOVA (**F**) or two-way ANOVA (**B**,**D**,**E**); * *p* < 0.05; ** *p* < 0.01; **** *p* < 0.0001.

**Figure 4 cancers-14-03769-f004:**
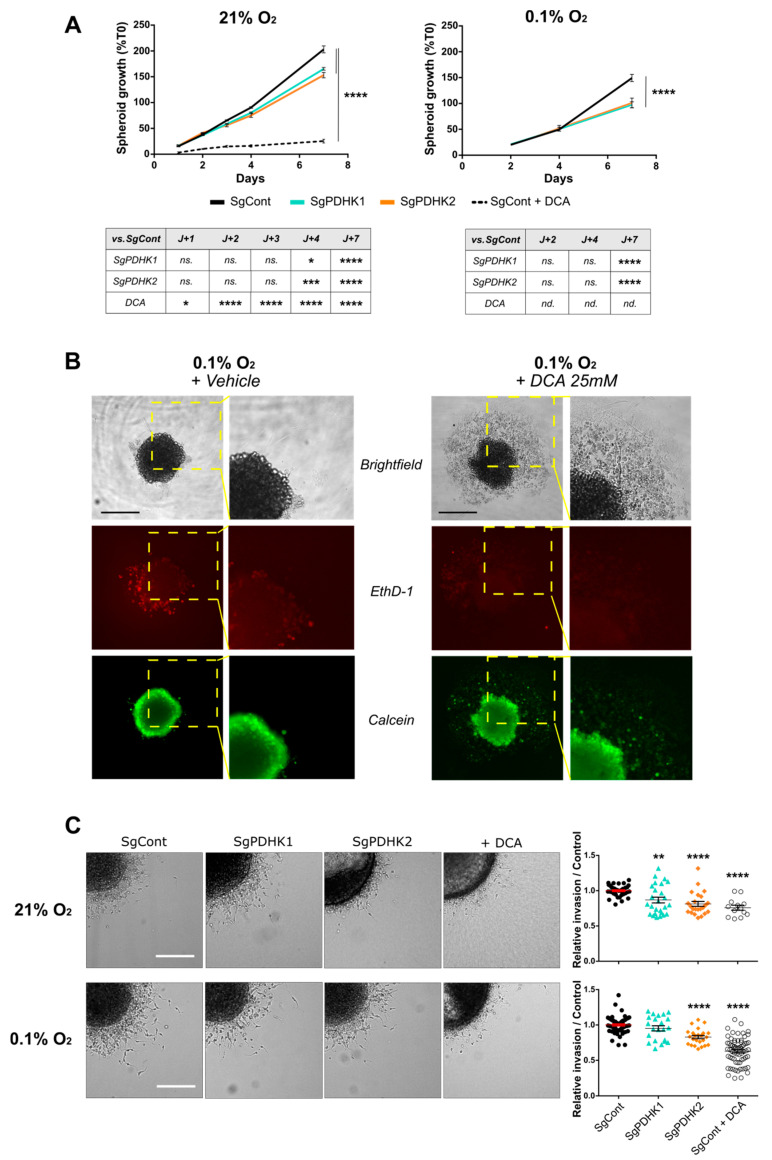
Genetic or pharmacologic PDHK disruption impairs GB cell proliferation and invasion. (**A**) *Left panel*: P3 spheroid growth recorded over 7 days in normoxic conditions (21% O_2_, 8–12 spheroids per condition, *n* = 3). Statistics versus control condition are presented in the table. *Right panel*: the same experiment was processed in hypoxic condition (0.1% O_2_). (**B**) Evaluation of live/dead cell status of spheroids exposed to both hypoxic conditions and DCA treatment. *Upper panel*: brightfield image of P3 SgCont spheroid before and after being exposed to vehicle or DCA 25 mM treatment for 7 days (left: vehicle; right: DCA). *Middle and lower panels*: viability assessment of spheroids, incubated with calcein (green) and ethidium homodimer (red), after a 7-day DCA or vehicle treatment. Scale bar: 150 µm. (**C**) *Left panel*: representative images of invasive spheroids SgCont, SgPDHK1, SgPDHK2 or treated with DCA, included in a collagen I matrix and incubated either at 21% or 0.1% O_2_ for 24 h. Scale bar: 150 µm. *Right panel*: relative invasion rate compared with the control condition (8–10 spheroids per condition, *n* = 3 to 5) upon 21% or 0.1% O_2_ exposure. Data are expressed as the mean ± SEM, and p-values were obtained using two-way ANOVA (**A**,**C**); * *p* < 0.05; ** *p* < 0.01; *** *p* < 0.001; **** *p* < 0.0001; *ns.*: not significant; *nd.*: not determined.

**Figure 5 cancers-14-03769-f005:**
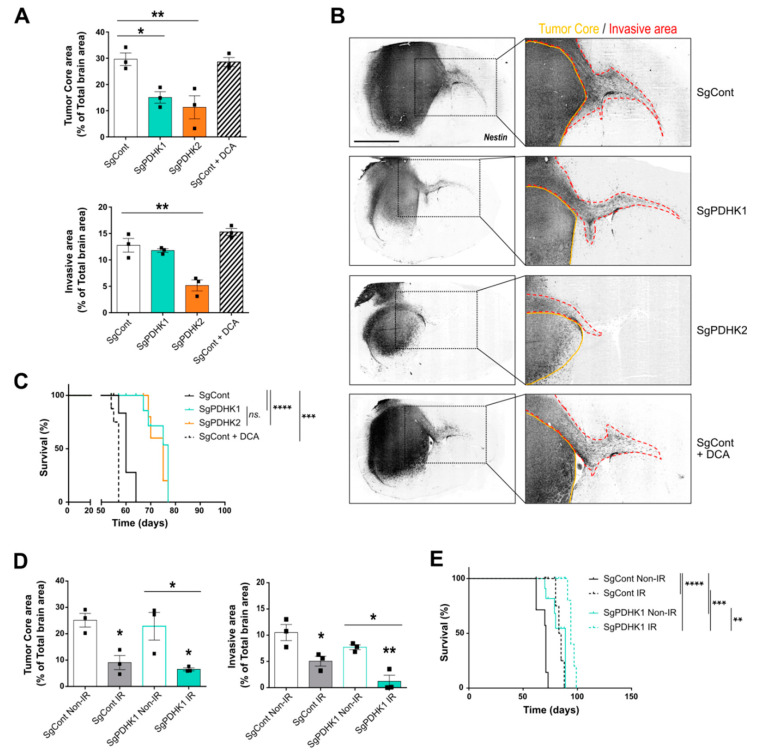
PDHK1 and PDHK2 KO reduce in vivo tumor growth and improve mice survival. (**A**) Tumor core and invasive areas were calculated in control, PDHK1 KO, PDHK2 KO and DCA-treated tumors based on mouse brain slices. (**B**) *Right panel:* histological analysis and staining with a human anti-nestin antibody (dark grey staining); *left panel:* magnified images showing contralateral invasion (red dashed lines). (**C**) Kaplan–Meier survival curves of xenotransplanted mice with P3 cells KO for PDHK1 (green), PDHK2 (orange), DCA-treated (dashed line) or control (black) (*n* = 5 to 8 mice per group). (**D**) Tumor core and invasive areas were calculated in control and PDHK1 KO tumors, irradiated (IR, 3 × 2 Gy) or not (Non-IR). (**E**) Kaplan–Meier survival curves of xenotransplanted mice with P3 cells KO for non-irradiated PDHK1 (green), or irradiated PDHK1 (green dashed line), or non-irradiated control (black) or irradiated control (black dashed line) (*n* = 4 to 7 mice per group). Data are expressed as the mean ± SEM, and p-values were obtained using one-way ANOVA (**A**,**D**) or log-rank (Mantel–Cox) test (**C**,**E**); * *p* < 0.05; ** *p* < 0.01; *** *p* < 0.001; **** *p* < 0.0001; *ns.*: not significant.

## Data Availability

The data that support the findings of this study are available on request from the corresponding author (TD).

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
