# Peer review of "Refining the Role of Pyruvate Dehydrogenase Kinases in Glioblastoma Development"

_cancers, 2022, doi:10.3390/cancers14153769_

Round 1
Reviewer 1 Report
Larrieu et al. investigated the distinct role of PDHK1 and PDHK2 in glioblastoma. The authors found that PDHK1 was enriched in the hypoxic area whereas PDHK2 was enriched in infiltrating tumor area. Furthermore, the authors showed that PDHK1 is mainly involved in energy metabolism and PDHK2 is involved in invasion, which correlates with their expression pattern in glioblastoma tissue. Also, knockout or inhibition of PDHK1 and PDHK2 impedes glioblastoma tumor growth and invasion.
1. In figure 2B-E, the authors showed the western blot result of 21% O2 and 0.1% O2 in separate membranes. However, the authors should show the results in the same blot to compare the expression level of protein (PDHK or PDH or p-PDH) in normoxic and hypoxic conditions. Also, in figure 2B and 2D, the authors should investigate the mRNA level of PDHK1 and 2 because hypoxic condition normally regulates the various protein level by modulating transcription.
2. In figure 2C, why is p-PDH increased in PDH1 knockout cells compared to control cells in 21% O2, though DCA treatment did not change PDHK1 protein level in figure 2D? Please explain it.
3. In lines 387-389, the authors explained single knockout of PDHK1 and PDHK2 did not change the sensitivity of cells to OXPHOS inhibitor due to the compensation effect between PDHKs. This explanation should be confirmed by examining the effect of the double knockdown of PDHK1 and PDHK2.
4. There is a discrepancy between in vitro data and in vivo data. DCA treatment showed a more apparent effect on inhibiting GBM cell growth and invasion compared to PDHK1 and PDHK 2 KO in vitro. However, DCA treatment barely suppressed the GBM cell growth and invasion in vivo. The authors should explain and discuss this discrepancy in the discussion section.
In figure 5D, why did the authors only show the tumor core area and invasive area in PDHK1 knockout mice? In previous results, PDHK2 knockout showed a better effect on inhibiting invasion. It would be better t
Author Response
Reviewer 1 :
Larrieu et al. investigated the distinct role of PDHK1 and PDHK2 in glioblastoma. The authors found that PDHK1 was enriched in the hypoxic area whereas PDHK2 was enriched in infiltrating tumor area. Furthermore, the authors showed that PDHK1 is mainly involved in energy metabolism and PDHK2 is involved in invasion, which correlates with their expression pattern in glioblastoma tissue. Also, knockout or inhibition of PDHK1 and PDHK2 impedes glioblastoma tumor growth and invasion.
Point 1-1. In figure 2B-E, the authors showed the western blot result of 21% O2 and 0.1% O2 in separate membranes. However, the authors should show the results in the same blot to compare the expression level of protein (PDHK or PDH or p-PDH) in normoxic and hypoxic conditions.
Reply: All western blots presented in Figure 2B-E show bands extracted from the same membrane with corresponding loading control (antibodies anti-vinculin) and representative quantification of all western blot replicates are presented below each panel. Thus, 21% and 0.1% O2 conditions presented in the same figure can be compared to each other. Full membranes and representative cropped areas were presented in supplementary data and are now better indicated in the text (lines 137-142).
Point 1-2. Also, in figure 2B and 2D, the authors should investigate the mRNA level of PDHK1 and 2 because hypoxic condition normally regulates the various protein level by modulating transcription.
Reply: Hypoxia is known to regulate PDHK1 expression, as there is a hypoxia-responsive element in the promoter of this gene (Kim et al., Cell Metab., 2006 - PMID: 16517405), and not in the promoter of PDHK2. In our RNAseq data from P3 cells (publication in revision doi: 10.21203/rs.3.rs-690811/v1), PDHK1 is upregulated 2.3-fold in log2FC between hypoxia and normoxia while PDHK2 gene expression is stable, results corresponding to protein expression shown by our western blot (Figure 2C).
Point 2. In figure 2C, why is p-PDH increased in PDH1 knockout cells compared to control cells in 21% O2, though DCA treatment did not change PDHK1 protein level in figure 2D? Please explain it.
Reply: Although no statistical difference was observed between conditions, we were surprised to detect an increased p-PDH in PDHK1 KO cells at 21% O2. We hypothesized that a knockout of PDHK1 could lead to a compensatory effect of other PDHK isoforms on the PDH phosphorylation as observed in Figure 2C. Indeed, PDHK2 has been shown to be more expressed in glioblastoma (Michelakis et al., Sci Transl med., 2010 - PMID: 20463368) than PDHK1 which has a low expression at 21% O2, as verified in the P3 cell model (Figure 2D). Moreover, in Figure 2D, we can see that DCA treatment at 21% O2 downregulates PDHK2 expression and consequently, a decrease in PDH phosphorylation is shown in Figure 2E. These data confirm that, at 21% O2, PDHK2 is probably the main kinase phosphorylating PDH, which would be more active when PDHK1 expression is suppressed, and that altering PDHK2 expression affects the total PDHK activity as shown by PDH phosphorylation. In hypoxic conditions, these regulations are different since PDHK1 expression is highly upregulated by low O2 levels, and takes the lead on PDH phosphorylation.
Point 3. In lines 387-389, the authors explained single knockout of PDHK1 and PDHK2 did not change the sensitivity of cells to OXPHOS inhibitor due to the compensation effect between PDHKs. This explanation should be confirmed by examining the effect of the double knockdown of PDHK1 and PDHK2.
Reply: In our study, we choose to use single knockouts of PDHK1 and PDHK2 in order to specifically analyze the role of each isoform. Moreover, as DCA partially inhibits PDHK1 and PDHK2 expression (as shown in our western blot experiments, Figure 2D) and fully impairs their activity (figure 2E, PDH phosphorylation), we hypothesize that DCA could mimic a double PDHK1/2 knockout in our cells. Furthermore, total inhibition of PDHK activity by DCA or genetic disruption of both PDHK1/PDHK2 expression should evenly impact PDH phosphorylation status and downstream metabolic pathways, such as cell respiration.
However, future experiments with a double PDHK1/2 knockout would reinforce our current data.
Point 4. There is a discrepancy between in vitro data and in vivo data. DCA treatment showed a more apparent effect on inhibiting GBM cell growth and invasion compared to PDHK1 and PDHK 2 KO in vitro. However, DCA treatment barely suppressed the GBM cell growth and invasion in vivo. The authors should explain and discuss this discrepancy in the discussion section.
Reply: As replied to the academic editor in Point 6, previous studies showed that DCA concentrations used in in vitro experiments vary from hundreds of micromolars to hundreds of millimolars, but also in in vivo experiments, from 2.5 mg/kg/day (Vella et al., Int J Cancer, 2012 - PMID: 21557214) to 200 mg/kg/day, the highest concentration for short term treatment (Lin et al., Br J Cancer, 2014 - PMID: 24892448). Furthermore, actual literature analyzing the effects of DCA in cancer is contradictory, showing alternatively positive anti-cancer effects but also low or no effect of DCA monotherapy, as reviewed by Stakisaitis et al. (Cancers, 2019 - PMID: 31434295).
In order to perform an initial experiment to evaluate the adapted dose of DCA for injection in Ragγ2C-/- mice, P3-implanted animals were treated with 25 mg/kg/day of DCA, which is the lowest dose found to be efficient in a neural cancer cell model (neuroblastoma) in mice (Vella et al., Int J Cancer, 2012 - PMID: 21557214). The lack of effect on tumor growth and invasion we observed in our in vivo experiment is certainly not due to cell-type specific response, but more probably to the stem cell status of the P3 cells. Indeed, in the literature where in vivo experiments were included, DCA has only been used on commercial glioblastoma cell lines known to be highly differentiated and mainly glycolytic (such as U87, U118, or U251 cells cultured with serum, Kumar et al., J Mol Med., 2013 - PMID: 23361368 ; Shen et al., Mol Cancer Ther., 2015 - PMID: 26063767). Thus, we hypothesized that a metabolic switch towards mitochondrial respiration induced by DCA is more effective in reducing tumor progression in these models when compared to the P3 stem-like cells, used in our study, that exhibit high metabolic adaptation capacities.
Surprisingly, the dose of DCA we used was significantly shortening survival in our experiment (p value < 0.0002 when compared to the control condition, and all DCA-treated mice reached endpoints in a 3-day timeframe), suggesting a toxic effect of DCA in our tumor model. Thus, regarding ethical purposes, we chose not to further increase DCA concentration in this model. However, as DCA effects were striking in in vitro experiments, combining PDHK inhibition with other therapies, such as temozolomide (alkylant agent used for glioblastoma therapy in patients) or radiations, will be considered as a new in vivo setup in further studies. Furthermore, a PDHK isoform-specific inhibitor could be more adapted to target specific areas of glioblastoma and disturb metabolic organization. These options are now raised in the discussion section of the revised manuscript.
Point 5. In figure 5D, why did the authors only show the tumor core area and invasive area in PDHK1 knockout mice? In previous results, PDHK2 knockout showed a better effect on inhibiting invasion. It would be better.
Reply: Both PDHK1 and PDHK2 knockout effects on tumor core and invasive area were evaluated in the first in vivo experiment we conducted, as presented in Figure 5A. Indeed, PDHK2 knockout tumors showed less invasive cells but clinical irradiation protocols efficiently target the tumor core and not the invasive cell population deeply disseminated in healthy brain parenchyma. In this context, to mirror clinical context and based on PDHK protein repartition in glioblastoma (PDHK1 is more expressed in tumor core, Figure 1B), we considered it more relevant to combine PDHK1 knockout and cranial irradiation to evaluate the impact of this combinatory approach in our glioblastoma model.
However, glioblastoma invasive cells are the roots of recurrent tumors and better targeting these cells appears central in glioblastoma therapy. Development of targeted strategies on invasive cells is one of the main projects of our lab and further studies will involve specific modulation of PDHK2 isoform, which is more expressed in the invasive population of glioblastoma (Figure 1D). These perspectives are now better discussed in the manuscript (lines 642-649).

Reviewer 2 Report
1. In this study, we have targeted cancer cell metabolism to disturb GB progression. GB???
2. Adding few recent citations from high impact journals defining PDKs in cancer shall be added to the introduction section https://doi.org/10.1016/j.bbcan.2021.188568
3. Dissociated cells were seeded in a 6-well plate and exposed to different experimental con ditions (treatments and/or different oxygen levels) for 48 h. Add relevant literature.
4. I found erratic usage of abbreviations at many places in the manuscript. Using of Grammarly software is advised.
5. Future perspective need to be added
6. The overall study is clinically significant carried out with sound methodology and the work is nicely presented and can be accepted after addressing these issues.
Author Response
Reviewer 2 :
- In this study, we have targeted cancer cell metabolism to disturb GB progression. GB???
Reply: This has been corrected in the text.
- Adding few recent citations from high impact journals defining PDKs in cancer shall be added to the introduction section https://doi.org/10.1016/j.bbcan.2021.188568
Reply: References have been added in the text and are in red in the text.
- Dissociated cells were seeded in a 6-well plate and exposed to different experimental conditions (treatments and/or different oxygen levels) for 48 h. Add relevant literature.
Reply: References have been added in the Material and Method section (Daubon et al., Nat Commun., 2019 - PMID: 30850588 and Guyon et al., doi: 10.21203/rs.3.rs-690811/v1).
- I found erratic usage of abbreviations at many places in the manuscript. Using of Grammarly software is advised.
Reply: We apologize for this issue. We have carefully read and corrected the text.
- Future perspective need to be added
Reply: We added future perspectives in the discussion section.
- The overall study is clinically significant carried out with sound methodology and the work is nicely presented and can be accepted after addressing these issues.
Reply: We would like to thank Reviewer 2 for these comments.

Round 2
Reviewer 1 Report
The points that were raised in the primary review have been well explained in the rebuttal letter.